# Learning Representations for Time Series Clustering

**Qianli Ma**
South China University of Technology
Guangzhou, China
qianlima@scut.edu.cn

**Jiawei Zheng**[*]
South China University of Technology
Guangzhou, China
csjwzheng@foxmail.com

**Sen Li** [*]
South China University of Technology
Guangzhou, China
awslee@foxmail.com

**Garrison W. Cottrell**
University of California, San Diego
CA, USA
gary@ucsd.edu

## Abstract

Time series clustering is an essential unsupervised technique in cases when category information is not available. It has been widely applied to genome data, anomaly detection, and in general, in any domain where pattern detection is important. Although feature-based time series clustering methods are robust to noise and outliers, and can reduce the dimensionality of the data, they typically rely on domain knowledge to manually construct high-quality features. Sequence to sequence (seq2seq) models can learn representations from sequence data in an unsupervised manner by designing appropriate learning objectives, such as reconstruction and context prediction. When applying seq2seq to time series clustering, obtaining a representation that effectively represents the temporal dynamics of the sequence, multi-scale features, and good clustering properties remains a challenge. How to best improve the ability of the encoder is still an open question. Here we propose a novel unsupervised temporal representation learning model, named Deep Temporal Clustering Representation (DTCR), which integrates the temporal reconstruction and K-means objective into the seq2seq model. This approach leads to improved cluster structures and thus obtains cluster-specific temporal representations. Also, to enhance the ability of encoder, we propose a fake-sample generation strategy and auxiliary classification task. Experiments conducted on extensive time series datasets show that DTCR is state-of-the-art compared to existing methods. The visualization analysis not only shows the effectiveness of cluster-specific representation but also shows the learning process is robust, even if K-means makes mistakes.

## 1 Introduction

Time series clustering is an important data mining technology widely applied to genome data [1], anomaly detection [2] and in general, to any domain where pattern detection is important. Time series clustering aids in the discovery of interesting patterns that empower data analysts to extract valuable information from complex and massive datasets [3].

Feature-based methods typically consist of extracted features and clusters. Such an approach is robust to noise and can filter out some irrelevant information [4], which can reduce the data dimension and thus improve the efficiency of clustering algorithms [3, 4]. However, most existing methods are domain-dependent, requiring domain knowledge to construct high-quality features manually [5]. In a

---

[*]Two authors have equal contribution.

number of studies [6, 7, 8, 9], discriminative features were selected with the help of pseudo cluster labels learned via local learning. However, the selected features are typically linear, while non-linear dynamics are more common in time series [10, 11, 12, 13].

In recent years, deep learning models have been applied to a wide variety of tasks and achieved great success. Among them, the seq2seq model can learn general representations from sequence data in an unsupervised manner by designing learning objectives that exploit labels that are freely available with the data [14]. For example, Kiros et al. [15] used it to learn the sentence representations by predicting the context sentences of a given sentence. Gan et al. [16] learned sentence representations by predicting multiple future sentences based on the seq2seq model. As shown by their experiments, if the general representations are fine-tuned using the downstream classification task, it can significantly improve the performance. This verifies the benefits of a task-related representation.

Motivated by this research, we aim to learn a non-linear temporal representation for time series clustering using the seq2seq model. When applying it to time series clustering, due to the lack of labels, effectively guiding the learning process to generate cluster-specific representations as well as capturing the dynamics and multi-scale characteristics of time series is a challenge. Moreover, the seq2seq model relies on the capabilities of the encoder. Improving the ability of the encoder for time series clustering remains an open question.

In this paper, we propose a novel unsupervised temporal representation learning model, Deep Temporal Clustering Representation (DTCR), which can generate cluster-specific temporal representations. DTCR integrates temporal reconstruction and the K-means objective into a seq2seq model. Specifically, DTCR adapts bidirectional Dilated recurrent neural networks [17] as the encoder, enabling the learned representation to capture the temporal dynamics and multi-scale characteristics of time series. The learned representation forms a cluster structure with the guidance of the K-means objective. To further enhance the ability of the encoder, inspired by [18], we propose a fake-sample generation strategy for time series and introduce an auxiliary classification task for the encoder. Our contributions can be summarized as follows:

1. We propose a novel unsupervised temporal representation learning model for time series clustering, which integrates the temporal reconstruction and K-means objective to generate cluster-specific temporal representations.

2. We propose a fake-sample generation strategy for time series and introduce an auxiliary classification task for the encoder to enhance its ability.

3. Our experimental results on a large number of benchmark time series datasets show that the proposed model achieves state-of-the-art performance. Visualization analysis illustrates the effectiveness of cluster-specific temporal representations and demonstrates the robustness of the learning process, even if K-means makes mistakes.

## 2  Related Work

Time series clustering algorithms can be broadly classified into two approaches: raw-data-based methods and feature-based methods [19].

### 2.1  Raw-data-based methods

Raw-data-based methods mainly modify the distance function to adapt to the time series characteristics (e.g., scaling and distortion). For example, Petitjean et al. [20] proposed a k-DBA algorithm for better alignment, which combines K-means and dynamic time warping [21]. Yang et al. [22] developed the K-Spectral Centroid (K-SC) method to uncover the temporal dynamics by using a similarity metric that is invariant to scaling and shifting. Paparrizos et al. [5] presented a method called k-Shape that further considers the shapes of the time series, using a normalized version of the cross-correlation measure. However, the above methods are usually sensitive to outliers and noise, since all time points are taken into account [4].

### 2.2  Feature-based methods

Feature-based methods use clustering algorithms on the extracted feature representations of input time series, which mitigates the impact of noise or outliers while also reducing the dimensionality of the

data. Since our method is related to this category, we here subdivide the feature-based methods into: (i) two-stage approaches that cluster after extracting features; (ii) approaches that jointly optimize the feature learning and clustering. The former first extracts features and then performs clustering. Guo et al. [23] used independent component analysis to convert the data into low dimensional features. Zakaria et al. [24] proposed u-shapelet to learn local patterns. The features extracted by these methods may not be suitable for clustering due to using the feature extraction as a pre-processing step. The latter category of algorithms jointly optimizes feature learning and clustering. In [6, 7, 8, 9], they iteratively adopted local learning to obtain pseudo-labels and then employed them to select discriminative features. The features extracted by these methods are linear, while real time series tend to be non-linear [10, 11, 12, 13]. Sai et al. [25] proposed deep temporal clustering (DTC), using an auto-encoder and a clustering layer [26] to learn a non-linear cluster representation. The clustering layer is designed by measuring the KL divergence between the predicted and target distribution. During training, the target distribution is calculated by the predicted distribution and updated at each iteration, which leads to instability [27]. Moreover, the performance of DTC strongly depends on the ability of the encoder since the predicted distribution is calculated on the learned representations.

# 3 Proposed Method

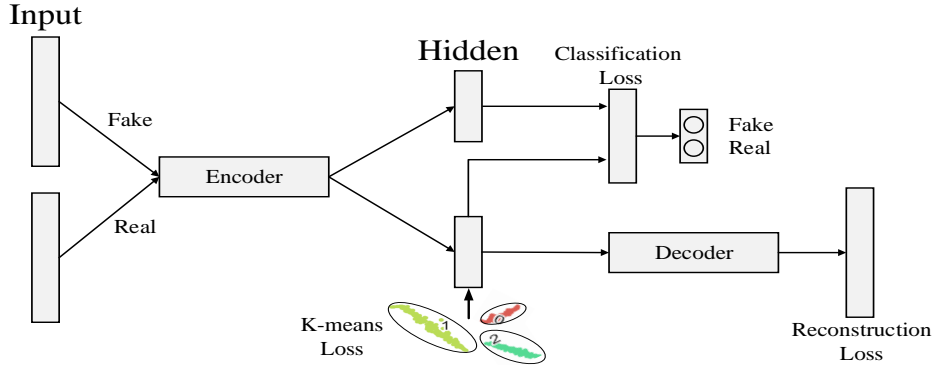

Figure 1: The general architecture of the Deep Temporal Clustering Representation (DTCR).

In this paper, we propose a novel model called Deep Temporal Clustering Representation(DTCR) to generate cluster-specific representations. The general structure of DTCR is illustrated in Figure 1. The encoder maps original time series into a latent space of representations. Then the representations are used to reconstruct the input data with the decoder. At the same time, a K-means objective is integrated into the model to guide the representation learning. Furthermore, we propose a fake-sample generation strategy and auxiliary classification task to enhance the ability of encoder.

## 3.1 Deep Temporal Representation Clustering

Given a set of $n$ time series $\boldsymbol{D} = \{\boldsymbol{x_1}, \boldsymbol{x_2}, ..., \boldsymbol{x_n}\}$, each time series $\boldsymbol{x_i}$ contains $T$ ordered real values denoted as $\boldsymbol{x_i} = (x_{i,1}, x_{i,2}, ...x_{i,T})$. Define non-linear mappings $f_{enc} : \boldsymbol{x_i} \rightarrow \boldsymbol{h_i}$ and $f_{dec} : \boldsymbol{h_i} \rightarrow \hat{\boldsymbol{x_i}}$. $f_{enc}$, $f_{dec}$, denotes the encoding and decoding process, respectively. $\boldsymbol{h_i} \in \mathbb{R}^m$ is the $m$-dimensional latent representation of time series $\boldsymbol{x_i}$, defined by:

$$\boldsymbol{h_i} = f_{enc}(\boldsymbol{x_i}) \tag{1}$$

We aim to train a good $f_{enc}$, making the learned representations facilitate the clustering task. We instantiate the non-linear mapping as a bidirectional RNN. Furthermore, considering that time series are commonly multi-scale, the encoder RNN is instantiated by a multi-layer Dilated RNN [17]. The latent representation is obtained by concatenating the last hidden state output of each layer of the Dilated RNN. After decoding, we can obtain the output $\hat{\boldsymbol{x_i}}$, where $\hat{\boldsymbol{x_i}} \in \mathbb{R}^T$ is given by:

$$\hat{\boldsymbol{x_i}} = f_{dec}(\boldsymbol{h_i}) \tag{2}$$

We use Mean Square Error (MSE) as the reconstruction loss, which is defined by:

$$\mathcal{L}_{reconstruction} = \frac{1}{n} \sum_{i=1}^{n} \| \boldsymbol{x_i} - \hat{\boldsymbol{x_i}} \|_2^2 \tag{3}$$

Note that although the learned representations by reconstruction loss capture the informative features of the original time series, they are not necessarily suitable for the clustering task. To enable the learned representations to form cluster structures and thus obtain cluster-specific representations, we further guide the network learning through k-means.

Given a static data matrix $\boldsymbol{H} \in \mathbb{R}^{m \times N}$, Zha et al. [28] showed that the minimization of K-means could be reformulated as a trace maximization problem associated with the Gram matrix $\boldsymbol{H^T H}$, which possesses optimal global solutions without local minima. Spectral relaxation converts the K-means objective into the following problem:

$$\mathcal{L}_{K-means} = Tr(\boldsymbol{H^T H}) - Tr(\boldsymbol{F^T H^T H F}) \tag{4}$$

where $Tr$ denotes the matrix trace. $\boldsymbol{F} \in \mathbb{R}^{N \times k}$ is the cluster indicator matrix. Considering $\boldsymbol{H}$ is given, the minimization of Eq. (4) can be further relaxed to a trace maximization problem by setting $\boldsymbol{F}$ to be an arbitrary orthogonal matrix:

$$\max_{\boldsymbol{F}} Tr(\boldsymbol{F^T H^T H F}), \ s.t. \ \boldsymbol{F^T F} = \boldsymbol{I} \tag{5}$$

The closed-form solution of $\boldsymbol{F}$ is obtained by composing the first $k$ singular vectors of $\boldsymbol{H}$ according to the Ky Fan theorem.

However, in our case, $\boldsymbol{H}$ is learned by the network instead of static. This motivates regarding Eq. (4) as a regularization term for learning $\boldsymbol{H}$, which guides the learning representation process, forming the cluster structures. Thus, our target is to minimize the objective below ($\lambda$ is a scalar):

$$\min_{\boldsymbol{H}, \boldsymbol{F}} J(\boldsymbol{H}) + \frac{\lambda}{2}[Tr(\boldsymbol{H^T H}) - Tr(\boldsymbol{F^T H^T H F})], \ s.t. \ \boldsymbol{F^T F} = \boldsymbol{I} \tag{6}$$

where $J(\boldsymbol{H})$ is the sum of the reconstruction loss and the classification loss (see Section 3.2). The whole training process of DTRC consists of iteratively updating $\boldsymbol{F}$ and $\boldsymbol{H}$. Fixing $\boldsymbol{F}$, updating $\boldsymbol{H}$ can follow the standard stochastic gradient descent (SGD), with the gradient given as: $\nabla J(\boldsymbol{H}) + \lambda \boldsymbol{H}(\boldsymbol{I} - \boldsymbol{F F^T})$. Fixing $\boldsymbol{H}$, we update $\boldsymbol{F}$ using the closed-form solution to Eq. (5), by computing the $k$-truncated singular value decomposition (SVD) of $\boldsymbol{H}$. In this way, the K-means objective guides the representations to form the cluster structures. Note that to avoid instability, $\boldsymbol{F}$ should not be updated at each iteration. In practice, we update $\boldsymbol{F}$ once after every 10 iterations. We provide an analysis of that in the section $F$ of the Supplementary Material.

### 3.2 Encoder Classification Task

Since the seq2seq model relies on the capabilities of the encoder, the better the encoder is trained, the better the learned representations will be. For time series, we propose a fake-sample generation strategy and auxiliary classification task to enhance the ability of the encoder.

Given a time series $\boldsymbol{x_i} \in \mathbb{R}^T$, we generate its fake version by randomly shuffling some time steps. The number of selected time steps is $\lfloor \alpha \times T \rfloor$, where $\alpha \in (0, 1]$ is a hyper-parameter we set to $0.2$. For each raw time series, we will generate the corresponding fake sample. The auxiliary classification task is to train the encoder to detect whether a given time series is real or fake. Formally, the encoder is trained by minimizing the following loss function:

$$\hat{\boldsymbol{y}_i} = \boldsymbol{W}_{fc2}(\boldsymbol{W}_{fc1}\boldsymbol{h_i}) \tag{7}$$

$$\mathcal{L}_{classification} = -\frac{1}{2N} \sum_{i=1}^{2N} \sum_{j=1}^{2} 1\{y_{i,j} = 1\} \log \frac{\exp \hat{y}_{i,j}}{\sum_{j=1}^{2} \exp(\hat{y}_{i,j})} \tag{8}$$

where $\boldsymbol{y_i}$ is a 2-dim one-hot vector indicating real or fake, and $\hat{\boldsymbol{y}_i}$ is the classification result. For simplicity, we ignore the bias term. $\boldsymbol{W}_{fc1} \in \mathbb{R}^{m \times d}$, $\boldsymbol{W}_{fc2} \in \mathbb{R}^{d \times 2}$ are parameters of the fully connected layers and $d$ is set to 128.

The ability of the encoder is enhanced to distinguish between real and fake samples, enabling the learned representation to better represent real time series.

---
**Algorithm 1** DTCR Training Method
---
**Input:** Data set: $D$; Number of clusters: $K$; Alternate update: $T$; Maximum iterations: $MaxIter$
**Output:** Cluster result $s$
 1: For each time series in $D$, generate the corresponding fake samples.
 2: **for** $iter = 1$ to $MaxIter$ **do**
 3:     Update latent representation $\{h_i = f_{enc}(x_i)\}_{i=1}^n$ using SGD based on Eq. (9).
 4:     **if** $iter \% T = 0$ **then**
 5:         Update $F$ using the closed-form solution of Eq. (5).
 6:     **end if**
 7: **end for**
 8: Apply K-means to the learned representation and get the cluster result $s$.
---

## 3.3 Overall Loss Function

Finally, the overall training loss $\mathcal{L}_{DTCR}$ of DTCR is defined by:

$$\mathcal{L}_{DTCR} = \mathcal{L}_{reconstruction} + \mathcal{L}_{classification} + \lambda\mathcal{L}_{K-means} \qquad (9)$$

where $\lambda$ is the regularization coefficient. Eq. (9) is minimized to learn the cluster-specific representations. Specifically, $\mathcal{L}_{reconstruction}$ makes the representations reconstruct the input. $\mathcal{L}_{classification}$ enhances the ability of the encoder. $\mathcal{L}_{K-means}$ encourages the representations to form cluster structures. After training, we apply K-means to the learned representations. The detailed training method of DTCR is presented in Algorithm 1.

## 4 Experiments

Following the protocol used in [20, 24, 5, 25, 29], we conduct experiments on the 36 UCR [30] time series datasets to evaluate performance. The statistics of these 36 datasets are shown in Table 1 of the Supplementary Material. Each data set has a default train/test split. We adopted the protocol used in USSL [29], training on the training set and evaluating on the test set for comparison. As mentioned above, we employ the bidirectional multi-layer Dilated RNN [17] as the encoder, capturing the dynamics and multi-scale characteristics of the time series. In our experiments, we fixed the number of layers and the number of dilation per layer to 3 and 1, 4, and 16, respectively. DTCR performs well under this setting. With further tuning, the performance could be improved. The decoder is a single-layer RNN. Gated Recurrent Units (GRU) are used in the RNNs [31]. The number of units per layer of the encoder is $[m_1, m_2, m_3] \subset \{[100, 50, 50], [50, 30, 30]\}$. The number of hidden units in the decoder is $(m_1 + m_2 + m_3) \times 2$. The decoder takes the final hidden state of the encoder as its initial state and performs iterative prediction, i.e., the output at time $t - 1$ is fed as the input at time $t$. The $\lambda$ of Eq. (9) $\in \{1, 1e - 1, 1e - 2, 1e - 3\}$. The batch size is $2N$. To reduce the impact of random initialization, we ran each experiment 5 times and report means and standard deviations.

The experiments are run on the TensorFlow [32] platform using an Intel Core i7 − 6850K, 3.60-GHz CPU, 64-GB RAM and a GeForce GTX 1080-Ti 11G GPU. The Adam [33] optimizer is employed with an initial learning rate of $5e - 3$.

### 4.1 Comparison with State-of-the-art Methods

Following USSL, the Rand Index [34] and Normalized Mutual Information [35] are used for evaluating clustering performance. The RI is defined as:

$$RI = \frac{TP + TN}{n(n-1)/2} \qquad (10)$$

where $TP$ (True Positive) is the number of pairs of time series that are correctly put in the same cluster, $TN$ (True Negative) is the number of pairs that are correctly put in different clusters and $n$ is the size of the data set.

The NMI is defined as:

$$NMI = \frac{\sum_{i=1}^{M}\sum_{j=1}^{M} N_{ij} \log(\frac{N \cdot N_{ij}}{|G_i||A_j|})}{\sqrt{(\sum_{i=1}^{M}|G_i|\log\frac{|G_i|}{N})(\sum_{j=1}^{M}|A_j|\log\frac{A_j}{N})}} \qquad (11)$$

where $N$ represents the total number of time series. $|G_i|$, $|A_j|$ are the number of time series in cluster $G_i$ and $A_j$. $N_{ij} = |G_i \bigcap A_j|$ denotes the number of time series belonging to the intersection of sets $G_i$ and $A_j$. In these two metrics, values close to 1 indicates high quality clustering [29].

We compare DTCR with 11 recently representative time series clustering methods. We also compare DTCR with 2 state-of-the-art non-time-series deep clustering methods (DEC [26], IDEC [27]). The details of these methods are described in section $B$ of the Supplementary Material. All the results in Table 1 are collected from [29] (2018 TPAMI) except for the new time series method DTC[2] [25] and two non time series methods (DEC[3], IDEC[4]). The results of these 3 methods are obtained by running their published code.

As shown in Table 1, DTCR achieves the best performance in terms of the lowest average rank of 3.0694, the highest average RI of 0.7714 and the number of best results 17. To further analyze the performance, we perform a pairwise comparison for each method against DTCR. Specifically, we conduct the Wilcoxon signed rank test [36] to measure the significance of the difference. As shown in Table 1, DTCR is significantly better than all of the other methods at $p < 0.05$ level, except USSL [29]. Although DTCR is numerically superior in average rank and RI, it is not significantly better than USSL. Note that USSL depends on pseudo-labels to guide the learning, while there is no mechanism to reduce the negative impact when mistakes occur in the pseudo-labels. In contrast, DTCR is capable of correcting mistakes with the help of temporal reconstruction (for analysis see Section 4.3.3). Due to space limitations, the results using the NMI metric are reported in Table 2 of the Supplementary Material. Note that DTCR also achieves the lowest average rank of 2.2500. We also show the performance on the ACC metric in Tables 3 and 4 of the Supplementary Material.

In addition, following the YADING paper [19], a larger and more complex dataset (StarLightCurves: 9236 samples, each sample's length is 1024) is used for evaluation. We adopted the same metric (NMI) for direct comparison. As shown in Table 2, DTCR again achieves the best performance.

Table 1: Rand Index (RI) comparisons on 36 time series datasets (the values in parentheses present standard deviations)

| Dataset | K-means [37] | UDFS [6] | NDFS [7] | RUFS [8] | RSFS [9] | KSC [22] | KDBA [20] | k-shape [5] | u-shapelet [24] | DTC [25] | USSL [29] | DEC [26] | IDEC [27] | DTCR |
|---|---|---|---|---|---|---|---|---|---|---|---|---|---|---|
| Arrow | 0.6905 | 0.7254 | 0.7381 | **0.7476** | 0.7108 | 0.7254 | 0.7222 | 0.7254 | 0.6460 | 0.6692 | 0.7159 | 0.5817 | 0.6210 | 0.6868(0.0026) |
| Beef | 0.6713 | 0.6759 | 0.7034 | 0.7149 | 0.6975 | 0.7057 | 0.6713 | 0.5402 | 0.6966 | 0.6345 | 0.6966 | 0.5954 | 0.6276 | **0.8046(0.0018)** |
| BeetleFly | 0.4789 | 0.4949 | 0.5579 | 0.6053 | 0.6516 | 0.6053 | 0.6052 | 0.6053 | 0.7314 | 0.5211 | 0.8105 | 0.4947 | 0.6053 | **0.9000(0.0001)** |
| BirdChicken | 0.4947 | 0.4947 | 0.7316 | 0.5579 | 0.6632 | 0.7316 | 0.6053 | 0.6632 | 0.5579 | 0.4947 | 0.8105 | 0.4737 | 0.4789 | **0.8105(0.0033)** |
| Car | 0.6345 | 0.6757 | 0.6260 | 0.6667 | 0.6708 | 0.6898 | 0.6254 | 0.7028 | 0.6418 | 0.6695 | 0.7345 | 0.6859 | 0.6870 | **0.7501(0.0022)** |
| chlorineConcentration | 0.5241 | 0.5282 | 0.5225 | 0.5330 | 0.5316 | 0.5256 | 0.5300 | 0.4111 | 0.5318 | 0.5353 | 0.4997 | 0.5348 | 0.5350 | **0.5357(0.0011)** |
| coffee | 0.7460 | 0.8624 | **1.0000** | 0.5476 | **1.0000** | **1.0000** | 0.4851 | **1.0000** | **1.0000** | 0.4841 | **1.0000** | 0.4921 | 0.5767 | 0.9286(0.0016) |
| diatomsizeReduction | 0.9583 | 0.9583 | 0.9583 | 0.9333 | 0.9137 | **1.0000** | 0.9583 | **1.0000** | 0.7083 | 0.8792 | **1.0000** | 0.9294 | 0.7347 | 0.9682(0.0032) |
| dist.phal.outl.agegroup | 0.6171 | 0.6531 | 0.6239 | 0.6252 | 0.6539 | 0.6535 | 0.6750 | 0.6020 | 0.6273 | 0.7812 | 0.6650 | 0.7785 | 0.7786 | **0.7825(0.0008)** |
| dist.phal.outl.correct | 0.5252 | 0.5362 | 0.5362 | 0.5252 | 0.5327 | 0.5235 | 0.5203 | 0.5252 | 0.5098 | 0.5010 | 0.5962 | 0.5029 | 0.5330 | **0.6075(0.0024)** |
| ECG200 | 0.6315 | 0.6533 | 0.6315 | 0.7018 | 0.6916 | 0.6315 | 0.6018 | 0.7018 | 0.5758 | 0.6018 | **0.7285** | 0.6422 | 0.6233 | 0.6648(0.0034) |
| ECGFiveDays | 0.4783 | 0.5020 | 0.5573 | 0.5020 | 0.5953 | 0.5257 | 0.5573 | 0.5020 | 0.5968 | 0.5016 | 0.8340 | 0.5103 | 0.5114 | **0.9638(0.0032)** |
| GunPoint | 0.4971 | 0.5029 | 0.5102 | 0.6498 | 0.4994 | 0.4971 | 0.5420 | 0.6278 | 0.6278 | 0.5400 | **0.7257** | 0.4981 | 0.4974 | 0.6398(0.0011) |
| Ham | 0.5025 | 0.5219 | 0.5362 | 0.5107 | 0.5127 | 0.5362 | 0.5141 | 0.5311 | 0.5362 | 0.5648 | **0.6393** | 0.5963 | 0.4956 | 0.5362(0.0035) |
| Herring | 0.4965 | 0.5099 | 0.5164 | 0.5238 | 0.5151 | 0.4940 | 0.5164 | 0.4965 | 0.5417 | 0.5045 | **0.6190** | 0.5099 | 0.5099 | 0.5759(0.0017) |
| Lighting2 | 0.4966 | 0.5119 | 0.5373 | 0.5729 | 0.5269 | 0.6263 | 0.5119 | 0.6548 | 0.5192 | 0.5770 | **0.6955** | 0.5311 | 0.5519 | 0.5913(0.0016) |
| Meat | 0.6595 | 0.6483 | 0.6635 | 0.6578 | 0.6657 | 0.6723 | 0.6816 | 0.6575 | 0.6742 | 0.3220 | 0.7740 | 0.6475 | 0.6220 | **0.9763(0.0016)** |
| Mid.phal.outl.agegroup | 0.5351 | 0.5269 | 0.5350 | 0.5315 | 0.5473 | 0.5364 | 0.5513 | 0.5105 | 0.5396 | 0.5757 | 0.5807 | 0.7059 | 0.6800 | **0.7982(0.0028)** |
| Mid.phal.outl.correct | 0.5000 | 0.5441 | 0.5047 | 0.5114 | 0.5149 | 0.5014 | 0.5063 | 0.5114 | 0.5218 | 0.5272 | **0.6635** | 0.5423 | 0.5423 | 0.5617(0.0006) |
| Mid.phal.TW | 0.0983 | 0.1225 | 0.1919 | 0.7920 | 0.8062 | 0.8187 | 0.8046 | 0.6213 | 0.7920 | 0.7115 | 0.7920 | 0.8590 | 0.8626 | **0.8638(0.0007)** |
| MoteStrain | 0.4947 | 0.5579 | 0.6053 | 0.5579 | 0.6168 | 0.6632 | 0.4789 | 0.6053 | 0.4789 | 0.5062 | **0.8105** | 0.7435 | 0.7324 | 0.7686(0.0036) |
| OSULeaf | 0.5615 | 0.5372 | 0.5622 | 0.5497 | 0.5665 | 0.5714 | 0.5541 | 0.5538 | 0.5525 | 0.7329 | 0.6551 | 0.7484 | 0.7607 | **0.7739(0.0014)** |
| Plane | 0.9081 | 0.8949 | 0.8954 | 0.9220 | 0.9314 | 0.9603 | 0.9225 | 0.9901 | **1.0000** | 0.9040 | **1.0000** | 0.9447 | 0.9447 | 0.9549(0.0037) |
| Prox.phal.outl.ageGroup | 0.5288 | 0.4997 | 0.5463 | 0.5780 | 0.5384 | 0.5305 | 0.5192 | 0.5617 | 0.5206 | 0.7430 | 0.7939 | 0.4263 | **0.8091** | 0.8091(0.0038) |
| Prox.phal.TW | 0.4789 | 0.4947 | 0.6053 | 0.5579 | 0.5211 | 0.6053 | 0.5211 | 0.5211 | 0.4789 | 0.8380 | 0.7282 | 0.8189 | **0.9030** | 0.9023(0.0023) |
| SonyAIBORobotSurface | 0.7721 | 0.7695 | 0.7721 | 0.7787 | 0.7928 | 0.7726 | 0.7988 | 0.8084 | 0.7639 | 0.5563 | 0.8105 | 0.5732 | 0.6900 | **0.8769(0.0033)** |
| SonyAIBORobotSurfaceII | 0.8697 | 0.8745 | 0.8865 | 0.8756 | 0.8948 | **0.9039** | 0.8684 | 0.5617 | 0.8770 | 0.7012 | 0.8575 | 0.6514 | 0.6572 | 0.8354(0.0016) |
| SwedishLeaf | 0.4987 | 0.4923 | 0.5500 | 0.5192 | 0.5038 | 0.4923 | 0.5500 | 0.5333 | 0.6154 | 0.8871 | 0.8547 | 0.8837 | 0.8893 | **0.9223(0.0021)** |
| Symbols | 0.8810 | 0.8548 | 0.8562 | 0.8525 | 0.9060 | 0.8982 | **0.9774** | 0.8373 | 0.9603 | 0.9053 | 0.9200 | 0.8841 | 0.8857 | 0.9168(0.0022) |
| ToeSegmentation1 | 0.4873 | 0.4921 | 0.5873 | 0.5429 | 0.4968 | 0.5000 | 0.6143 | 0.6143 | 0.5873 | 0.5077 | **0.6718** | 0.4984 | 0.5017 | 0.5659(0.0006) |
| ToeSegmentation2 | 0.5257 | 0.5257 | 0.5968 | 0.5968 | 0.5826 | 0.5257 | 0.5573 | 0.5257 | 0.5020 | 0.5348 | 0.6778 | 0.4991 | 0.4991 | **0.8286(0.0028)** |
| TwoPatterns | 0.8529 | 0.8259 | 0.8530 | 0.8385 | **0.8588** | 0.8585 | 0.8446 | 0.8046 | 0.7757 | 0.6251 | 0.8318 | 0.6293 | 0.6338 | 0.6984(0.0025) |
| TwoLeadECG | 0.5476 | 0.5495 | 0.6328 | 0.8246 | 0.5635 | 0.5464 | 0.5476 | 0.8246 | 0.5404 | 0.5116 | **0.8628** | 0.5007 | 0.5016 | 0.7114(0.0014) |
| wafer | 0.4925 | 0.4925 | 0.5263 | 0.5263 | 0.4925 | 0.4925 | 0.4925 | 0.4925 | 0.4925 | 0.5324 | **0.8246** | 0.5679 | 0.5597 | 0.7338(0.0006) |
| Wine | 0.4984 | 0.4987 | 0.5123 | 0.5021 | 0.5033 | 0.5006 | 0.5064 | 0.5001 | 0.5033 | 0.4906 | **0.8985** | 0.4913 | 0.5157 | 0.6271(0.0039) |
| WordsSynonyms | 0.8775 | 0.8697 | 0.8760 | 0.8861 | 0.8817 | 0.8727 | 0.8159 | 0.7844 | 0.8230 | 0.8855 | 0.8540 | 0.8893 | 0.8947 | **0.8984(0.0003)** |
| AVG Rank | 10.6667 | 9.6806 | 7.2222 | 7.3889 | 6.8750 | 7.1389 | 7.9167 | 8.2361 | 8.2500 | 8.8194 | 3.5000 | 8.6528 | 7.5833 | **3.0694** |
| AVG RI | 0.5975 | 0.6077 | 0.6402 | 0.6478 | 0.6542 | 0.6582 | 0.6335 | 0.6419 | 0.6402 | 0.6238 | 0.7676 | 0.6351 | 0.6515 | **0.7714** |
| Best | 0 | 0 | 1 | 1 | 1 | 3 | 2 | 0 | 1 | 0 | 12 | 0 | 1 | **17** |
| p-value | 2.089E-6 | 4.8823E-6 | 3.4131E-5 | 5.7729E-5 | 4.1222E-5 | 1.3545E-4 | 1.2565E-5 | 1.4814E-4 | 3.4141E-5 | 3.0287E-7 | 9.7386E-1 | 8.7697E-07 | 3.2916E-7 | - |

## 4.2 Ablation Study

To verify the effectiveness of the $\mathcal{L}_{K-means}$ and $\mathcal{L}_{classification}$, here we show a comparison between the full DTCR model and its two ablation models: 1) DTCR without K-means loss; and 2) DTCR

[2]https://github.com/saeeeeru/dtc-tensorflow

[3]https://github.com/piiswrong/dec

[4]https://github.com/XifengGuo/IDEC

Table 2: Normalized Mutual Information (NMI) comparisons on StarLightCurves

| Dataset | YADING | DEC | IDEC | DTC | DTCR |
|---|---|---|---|---|---|
| StarLightCurves | 0.6000 | 0.6058 | 0.6056 | 0.6072 | **0.6731** |

Table 3: Rand Index (RI) ablation study results of DTCR

| No. | Dataset | w/o K-means | w/o classification | DTCR | No. | Dataset | w/o K-means | w/o classification | DTCR |
|---|---|---|---|---|---|---|---|---|---|
| 1 | Arrow | 0.5980 | 0.5698 | **0.6868** | 19 | Mid.phal.outl.correct | 0.5137 | 0.5033 | **0.5617** |
| 2 | Beef | 0.7352 | 0.6497 | **0.8046** | 20 | Mid.phal.TW | 0.8625 | 0.8620 | **0.8638** |
| 3 | BeetleFly | 0.6305 | 0.6053 | **0.9000** | 21 | MoteStrain | 0.7121 | 0.7239 | **0.7686** |
| 4 | BirdChicken | 0.5600 | 0.4821 | **0.8105** | 22 | OSULeaf | 0.7416 | 0.7314 | **0.7739** |
| 5 | Car | 0.6610 | 0.6688 | **0.7501** | 23 | Plane | 0.9530 | 0.9409 | **0.9549** |
| 6 | chlorineConcentration | 0.5341 | 0.5004 | **0.5357** | 24 | Prox.phal.outl.ageGroup | 0.8004 | 0.7922 | **0.8091** |
| 7 | coffee | 0.6672 | 0.5434 | **0.9286** | 25 | Prox.phal.TW | 0.8549 | 0.8359 | **0.9023** |
| 8 | diatomsizeReduction | 0.8892 | 0.7851 | **0.9682** | 26 | SonyAIBORobotSurface | 0.7561 | 0.7702 | **0.8769** |
| 9 | dist.phal.outl.agegroup | 0.7775 | 0.7780 | **0.7825** | 27 | SonyAIBORobotSurfaceII | 0.7069 | 0.6332 | **0.8354** |
| 10 | dist.phal.outl.correct | 0.5056 | 0.5051 | **0.6075** | 28 | SwedishLeaf | 0.9107 | 0.9047 | **0.9223** |
| 11 | ECG200 | 0.6064 | 0.5412 | **0.6648** | 29 | Symbols | 0.8989 | 0.9043 | **0.9168** |
| 12 | ECGFiveDays | 0.6970 | 0.5623 | **0.9638** | 30 | ToeSegmentation1 | 0.5598 | 0.4993 | **0.5659** |
| 13 | GunPoint | 0.5589 | 0.4969 | **0.6398** | 31 | ToeSegmentation2 | 0.6878 | 0.6012 | **0.8286** |
| 14 | Ham | 0.5330 | 0.5040 | **0.5362** | 32 | TwoPatterns | 0.6537 | 0.6650 | **0.6984** |
| 15 | Herring | 0.5173 | 0.4967 | **0.5759** | 33 | TwoLeadECG | 0.5316 | 0.5262 | **0.7114** |
| 16 | Lighting2 | 0.5626 | 0.5554 | **0.5913** | 34 | wafer | 0.5900 | 0.5322 | **0.7338** |
| 17 | Meat | 0.8245 | 0.7181 | **0.9763** | 35 | Wine | 0.5642 | 0.5159 | **0.6271** |
| 18 | Mid.phal.outl.agegroup | 0.7981 | 0.7923 | **0.7982** | 36 | WordsSynonyms | 0.8920 | 0.8891 | **0.8984** |

without the auxiliary classification loss (w/o classification loss). Table 3 shows that the full DTCR is always superior to all of its ablations, demonstrating the effectiveness of the $\mathcal{L}_{K-means}$ and $\mathcal{L}_{classification}$.

### 4.3 Visualization Analysis

Through visualization, we analyze the benefits of the cluster-specific representations and illustrate the robustness of our model even if K-means makes mistakes. In all of the following experiments, we use t-SNE [38] to map the learned representations into $2D$ and plot it.

#### 4.3.1 Contribution of Each Loss

To explore the effectiveness of cluster-specific representations, we visualize the representations learned by DTCR and two of its ablations on datasets *ECGFiveDays* and *SonyAIBORobotSurface*. As shown in Figure 2, it is obvious that the representations learned by DTCR have formed 2 clusters despite a small amount of mixing. In contrast, the results of DTCR without K-means loss presents no cluster shape, and contain a small amount of mixing as well. As for DTCR without classification loss, the representations are also mixed, which verifies the importance of the ability of the encoder.

#### 4.3.2 The Process of Learning Representations

To better understand how DTCR learns the cluster-specific representations, we visualize its learning process. As shown in Figure 3, the representations at the beginning are scattered and chaotic. At Epoch 30, the prototype of 2 clusters has been formed. At Epoch 50, a well learned cluster-specific representation is established in terms of the small distance of intra-class and the large inter-class distance. This experiment has been conducted on all other datasets and the same experimental results were obtained. We report these in Table 3 of the Supplementary Material.

#### 4.3.3 Robustness Analysis

Since our model uses the information provided by K-means, what if K-means makes mistakes? Here we argue that our model is capable of correcting mistakes with the help of $\mathcal{L}_{reconstruction}$ and verify this point with some experiments.

Note that DTCR has 3 loss terms: $\mathcal{L}_{reconstruction}$, $\mathcal{L}_{K-means}$ and $\mathcal{L}_{classification}$. We can disrupt $\mathcal{L}_{K-means}$ while retaining either $\mathcal{L}_{reconstruction}$ or $\mathcal{L}_{classification}$ to figure out which one takes a more important role in preventing being misled by K-means. Here are the detailed settings: First, DTCR is trained with all loss terms for $50$ epochs, and we plot the learned representations as the initial

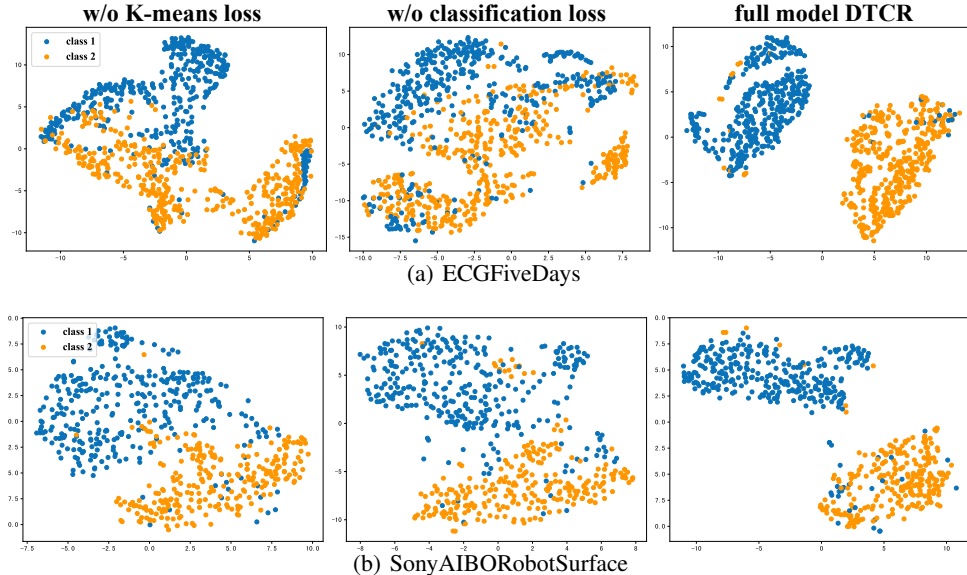

(a) ECGFiveDays

(b) SonyAIBORobotSurface

Figure 2: The visualizations with t-SNE on the datasets (a) *ECGFiveDays* and (b) *SonyAIBORobot-Surface*. The colors of the points indicate the actual labels.

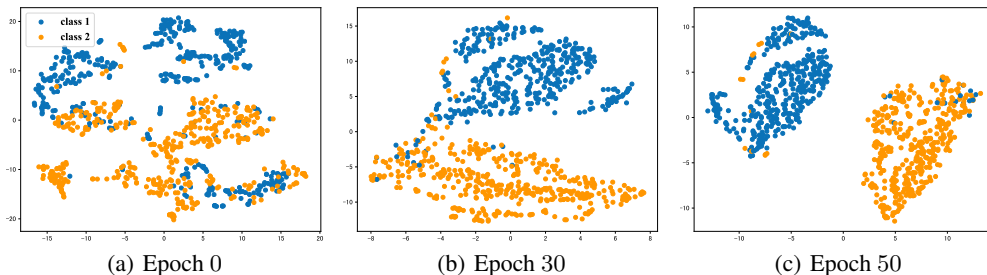

(a) Epoch 0         (b) Epoch 30         (c) Epoch 50

Figure 3: The learned representations on data set *ECGFiveDays* during the training process. From left to the right, the subfigure is obtained at Epoch 0, 30 and 50, respectively.

state. Then, we randomly shuffle the clustered index matrix $F$ (disrupting the term of $\mathcal{L}_{K-means}$) while retaining only one term of $\mathcal{L}_{reconstruction}$ or $\mathcal{L}_{classification}$, training for 50 epochs. We plot the learned representation as an intermediate state. Finally, we put the missing loss term back and train DTCR with all loss terms for another 50 epochs, obtaining the final state.

As shown in the first row of Figure. 4, when we train the DTCR with only the loss term of shuffled K-means and the classification, the wrong clustering information does mislead the learning, decreasing RI and thus the representations are mixed (Fig. 4 (b)). However, once adding $\mathcal{L}_{reconstruction}$ back, the RI is improved, indicating the learning of the model was corrected (Fig. 4 (c)). Similarly, we do that again to check what happens without $\mathcal{L}_{classification}$. As the second row of Figure. 4 shows, even without $\mathcal{L}_{classification}$ but with the help of $\mathcal{L}_{reconstruction}$, the RI is still improved and less confused (Fig. 4 (e)). Finally, putting the $\mathcal{L}_{classification}$ back improves the RI. Comparing Fig. 4 (b) with (e), it is clear that $\mathcal{L}_{reconstruction}$ enables our model to correct mistakes. Comparing Fig. 4 (c) with (f) shows that the earlier and longer the $\mathcal{L}_{reconstruction}$ is used, the stronger the ability to prevent being misled by K-means and thus obtaining the higher RI. Note that $\mathcal{L}_{reconstruction}$ is only put back in Fig. 4 (c) (trained for 50 epochs) while it is used in Fig. 4 (e) and (f) (trained for 100 epochs). The robustness analysis has been conducted on all other datasets and shows the same results (See the details in Section $E$ of the Supplementary Material).

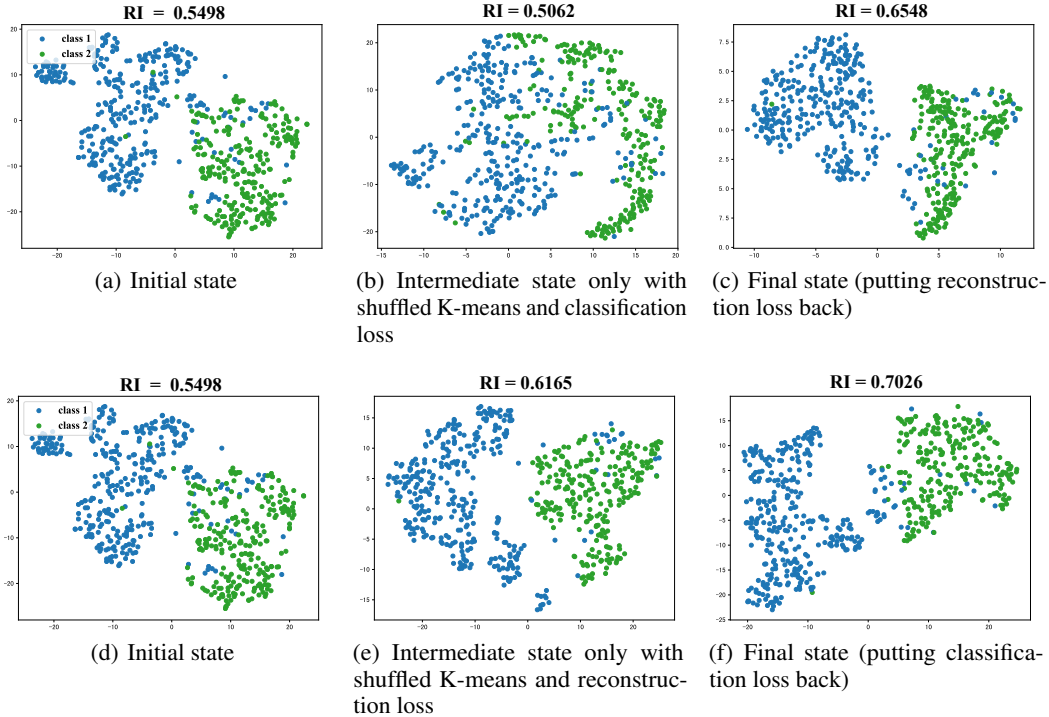

Figure 4: Robustness Analysis of DTCR on *SonyAIBORobotSurface*. Note that the (d) is the same as (a), replicated here for better illustration; hence the first and second rows start with the same state.

# 5    Conclusion

In this paper, we propose a novel model called Deep Temporal Clustering Representation (DTCR) that effectively generates cluster-specific representations. We integrate the temporal reconstruction and K-means objective into the seq2seq model, enabling the learned representations to encode the time series and to form cluster structures. Moreover, a fake-sample generation strategy for time series and auxiliary classification task are proposed to enhance the ability of the encoder. The extensive experimental results verify the effectiveness of the proposed method. Furthermore, we provide the visualization analysis to demonstrate the advantages of the cluster-specific representations and show the learning process is robust even if K-means makes mistakes. How to extend our clustering framework to time series with missing values is left for future work.

**Acknowledgments**

We thank the anonymous reviewers for their helpful feedbacks. The work described in this paper was partially funded by the National Natural Science Foundation of China (Grant Nos. 61502174, 61872148), the Natural Science Foundation of Guangdong Province (Grant Nos. 2017A030313355, 2017A030313358, 2019A1515010768), the Guangzhou Science and Technology Planning Project (Grant Nos. 201704030051, 201902010020).

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
