[Supplementary Material]

# Supplementary Material: Learning Representations for Time Series Clustering

**Qianli Ma**
South China University of Technology
Guangzhou, China
qianlima@scut.edu.cn

**Jiawei Zheng**[*]
South China University of Technology
Guangzhou, China
csjwzheng@foxmail.com

**Sen Li** [*]
South China University of Technology
Guangzhou, China
awslee@foxmail.com

**Garrison W. Cottrell**
University of California, San Diego
CA, USA
gary@ucsd.edu

## A    Dataset Introduction

In Section 4, we report the experimental results of DTCR on 36 UCR datasets [1]. Here, we show the statistics of these 36 datasets.

Table 1: Statistics of the benchmark time series datasets

| No. | Dataset | #Train/Test | Length | #classes | No. | Dataset | #Train/Test | Length | #classes |
|---|---|---|---|---|---|---|---|---|---|
| 1 | Arrow | 36/175 | 252 | 3 | 19 | Mid.phal.outl.correct | 291/600 | 81 | 2 |
| 2 | Beef | 30/30 | 471 | 5 | 20 | Mid.phal.TW | 154/399 | 81 | 6 |
| 3 | BeetleFly | 20/20 | 513 | 2 | 21 | MoteStrain | 20/1252 | 85 | 2 |
| 4 | BirdChicken | 20/20 | 513 | 2 | 22 | OSULeaf | 200/242 | 428 | 6 |
| 5 | Car | 60/60 | 578 | 4 | 23 | Plane | 105/105 | 145 | 7 |
| 6 | ChlorineConcentration | 467/3840 | 167 | 3 | 24 | Prox.phal.outl.ageGroup | 400/205 | 81 | 3 |
| 7 | Coffee | 28/28 | 287 | 2 | 25 | Prox.phal.TW | 205/400 | 81 | 6 |
| 8 | DiatomsizeReduction | 16/306 | 346 | 4 | 26 | SonyAIBORobotSurface | 20/601 | 71 | 2 |
| 9 | Dist.phal.outl.agegroup | 139/400 | 81 | 3 | 27 | SonyAIBORobotSurfaceII | 27/953 | 66 | 2 |
| 10 | Dist.phal.outl.correct | 276/600 | 81 | 2 | 28 | SwedishLeaf | 500/625 | 129 | 15 |
| 11 | ECG200 | 100/100 | 97 | 2 | 29 | Symbols | 25/995 | 399 | 6 |
| 12 | ECGFiveDays | 23/861 | 137 | 2 | 30 | ToeSegmentation1 | 40/228 | 278 | 2 |
| 13 | GunPoint | 50/150 | 151 | 2 | 31 | ToeSegmentation2 | 36/130 | 344 | 2 |
| 14 | Ham | 109/105 | 432 | 2 | 32 | TwoPatterns | 1000/4000 | 129 | 4 |
| 15 | Herring | 64/64 | 513 | 2 | 33 | TwoLeadECG | 23/1139 | 83 | 2 |
| 16 | Lighting2 | 60/61 | 638 | 2 | 34 | Wafer | 1000/6164 | 153 | 2 |
| 17 | Meat | 60/60 | 449 | 3 | 35 | Wine | 57/54 | 235 | 2 |
| 18 | Mid.phal.outl.agegroup | 154/400 | 81 | 3 | 36 | WordsSynonyms | 267/638 | 271 | 25 |

## B    Details of Baseline Methods

We compare DTCR with 11 recently representative time series clustering methods and 2 state-of-the-art non time series deep clustering methods (DEC [2], IDEC [3]). The details are as follows:

- K-means: Use K-means on the entire time series.
- UDFS [4]: Unsupervised discriminative feature selection that simultaneously explores the manifold structure, local discriminative information, and feature correlations.
- NDFS [5]: Non-negative discriminative feature selection that adopts $l_{2,1}$ regularised regression and non-negative spectral analysis as a joint framework for selecting features.

---

[*]Two authors have equal contribution.

- RUFS [6]: Robust unsupervised feature selection that uses robust orthogonal non-negative matrix factorization to perform feature learning jointly.

- RSFS [7]: Robust spectral learning for unsupervised feature selection, which joins spectral regression with sparse graph embedding.

- KSC [8]: Uses K-means for clustering by adopting a pairwise scaling distance measure and computing the spectral norm of a matrix for centroid computation.

- k-DBA [9]: Adopts K-means and dynamic time warping distance to obtain centroids, via a DBA method.

- k-shape [10]: Adopts a scalable iterative refinement procedure to explore the shapes of time series that have a normalized cross-correlation measure.

- u-shapelet [11]: A time series clustering method that deliberately ignores the rest of the data and only uses local patterns to cluster the time series.

- DTC [12]: Takes the KL divergence between predicted and target distribution as guidance to learn non-linear features in a deep framework.

- USSL [13]: Integrates the strengths of shapelet learning, shapelet regularization, spectral analysis, and pseudo-labels to help to cluster unlabeled time series better.

- DEC [2]: Learns a mapping from the data space to a lower-dimensional feature space in which it iteratively optimizes a clustering objective.

- IDEC [3]: Manipulates feature space to scatter data by optimizing a KL divergence-based clustering loss and maintains the local structure carefully.

## C   Comparison with State-of-the-art Methods

Table 2 reports the metric Normalized mutual information (NMI) of each algorithm on the 36 UCR time series datasets. The best performance for each dataset is highlighted in bold. And our method also achieves the lowest average rank of 2.2500 while USSL achieve 2.3472 on the NMI metric.

Table 2: Normalized Mutual Information (NMI) comparisons on 36 time series datasets (the values in parentheses represent standard deviations)

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

# D    The Process of Learning Representation

In section $4.3.2$ (The Process of Learning Representation) of the main text, we visualize the learning process of DTCR on 2 specific datasets, demonstrating that DTCR is capable of learning the cluster-specific representations as the iterative process proceeds. Here we report the experimental results on all other datasets by recording the improvements on the metrics. As shown in Table 3, the performance is gradually improving during the learning process (indicating by an upward arrow) demonstrating that the learned representations are more and more suitable for clustering.

Table 3: The improvements of the performance during the training process

| Dataset | #Epoch 0 (RI/NMI/ACC) | #Epoch 30 (RI/NMI/ACC) | #Epoch 50 (RI/NMI/ACC) |
|---|---|---|---|
| Arrow | 0.4126 / 0.3317 / 0.4152 | 0.4952 / 0.4119 / 0.5137↑ | 0.5717 / 0.4687 / 0.5796↑ |
| Beef | 0.4828 / 0.3371 / 0.3406 | 0.5942 / 0.4025 / 0.4205↑ | 0.6778 / 0.4650 / 0.4815↑ |
| BeetleFly | 0.5403 / 0.4566 / 0.5105 | 0.6639 / 0.5640 / 0.6213↑ | 0.7640 / 0.6436 / 0.7115↑ |
| BirdChicken | 0.4876 / 0.3189 / 0.5100 | 0.5980 / 0.3911 / 0.6182↑ | 0.6870 / 0.4483 / 0.7011↑ |
| Car | 0.4503 / 0.3023 / 0.3798 | 0.5532 / 0.3676 / 0.4691↑ | 0.6235 / 0.4244 / 0.5337↑ |
| chlorineConcentration | 0.3216 / 0.0279 / 0.5368 | 0.3996 / 0.0333 / 0.6688↑ | 0.4632 / 0.0385 / 0.7572↑ |
| coffee | 0.5600 / 0.4892 / 0.5784 | 0.6605 / 0.6055 / 0.7185↑ | 0.7665 / 0.7098 / 0.8239↑ |
| diatomsizeReduction | 0.5837 / 0.5666 / 0.5475 | 0.7059 / 0.6864 / 0.6614↑ | 0.8235 / 0.7810 / 0.7517↑ |
| dist.phal.outl.agegroup | 0.4701 / 0.2730 / 0.4832 | 0.5801 / 0.3379 / 0.5910↑ | 0.6614 / 0.3870 / 0.6830↑ |
| dist.phal.outl.correct | 0.3658 / 0.0707 / 0.4249 | 0.4526 / 0.0851 / 0.5163↑ | 0.5118 / 0.0973 / 0.5788↑ |
| ECG200 | 0.3987 / 0.2218 / 0.4825 | 0.4962 / 0.2679 / 0.5924↑ | 0.5557 / 0.3121 / 0.6888↑ |
| ECGFiveDays | 0.5783 / 0.4856 / 0.5123 | 0.7029 / 0.5874 / 0.6160↑ | 0.8035 / 0.6802 / 0.7196↑ |
| GunPoint | 0.3839 / 0.2526 / 0.4719 | 0.4712 / 0.3137 / 0.5755↑ | 0.5342 / 0.3562 / 0.6645↑ |
| Ham | 0.3222 / 0.0592 / 0.3829 | 0.3961 / 0.0730 / 0.4735↑ | 0.4582 / 0.0831 / 0.5407↑ |
| Herring | 0.3459 / 0.1354 / 0.4220 | 0.4191 / 0.1690 / 0.5220↑ | 0.4869 / 0.1965 / 0.5885↑ |
| Lighting2 | 0.3555 / 0.1375 / 0.4330 | 0.4284 / 0.1640 / 0.5244↑ | 0.4900 / 0.1923 / 0.6150↑ |
| Meat | 0.5863 / 0.5797 / 0.5872 | 0.7254 / 0.7145 / 0.7261↑ | 0.8190 / 0.8185 / 0.8213↑ |
| Mid.phal.outl.agegroup | 0.4792 / 0.2799 / 0.4619 | 0.5763 / 0.3474 / 0.5719↑ | 0.6508 / 0.4034 / 0.6380↑ |
| Mid.phal.outl.correct | 0.3369 / 0.0689 / 0.4040 | 0.4110 / 0.0841 / 0.4886↑ | 0.4837 / 0.0954 / 0.5483↑ |
| Mid.phal.TW | 0.5183 / 0.3303 / 0.3568 | 0.6316 / 0.3965 / 0.4394↑ | 0.7195 / 0.4597 / 0.4996↑ |
| MoteStrain | 0.4611 / 0.2459 / 0.5036 | 0.5605 / 0.3009 / 0.6206↑ | 0.6492 / 0.3458 / 0.7159↑ |
| OSULeaf | 0.4645 / 0.1563 / 0.2658 | 0.5747 / 0.1969 / 0.3202↑ | 0.6628 / 0.2235 / 0.3716↑ |
| Plane | 0.5752 / 0.5585 / 0.3781 | 0.7034 / 0.7019 / 0.4766↑ | 0.8019 / 0.8039 / 0.5378↑ |
| Prox.phal.outl.ageGroup | 0.4895 / 0.3450 / 0.4684 | 0.5895 / 0.4256 / 0.5713↑ | 0.6641 / 0.4753 / 0.6562↑ |
| Prox.phal.TW | 0.5416 / 0.4186 / 0.3782 | 0.6594 / 0.5151 / 0.4517↑ | 0.7607 / 0.5885 / 0.5139↑ |
| SonyAIBORobotSurface | 0.5264 / 0.4597 / 0.5570 | 0.6391 / 0.5627 / 0.6731↑ | 0.7238 / 0.6562 / 0.7735↑ |
| SonyAIBORobotSurfaceII | 0.5015 / 0.3699 / 0.5435 | 0.6269 / 0.4574 / 0.6540↑ | 0.7025 / 0.5243 / 0.7726↑ |
| SwedishLeaf | 0.5570 / 0.4017 / 0.2289 | 0.6731 / 0.5034 / 0.2697↑ | 0.7699 / 0.5753 / 0.3119↑ |
| Symbols | 0.5520 / 0.5395 / 0.3901 | 0.6723 / 0.6656 / 0.4881↑ | 0.7808 / 0.7470 / 0.5503↑ |
| ToeSegmentation1 | 0.3396 / 0.1869 / 0.3395 | 0.4145 / 0.2338 / 0.4151↑ | 0.4736 / 0.2633 / 0.4684↑ |
| ToeSegmentation2 | 0.4988 / 0.2340 / 0.5408 | 0.6084 / 0.2841 / 0.6457↑ | 0.7065 / 0.3243 / 0.7477↑ |
| TwoPatterns | 0.4210 / 0.2836 / 0.2555 | 0.5099 / 0.3470 / 0.3155↑ | 0.5987 / 0.3992 / 0.3641↑ |
| TwoLeadECG | 0.4279 / 0.2770 / 0.5178 | 0.5059 / 0.3419 / 0.6290↑ | 0.5814 / 0.3883 / 0.7188↑ |
| wafer | 0.4420 / 0.0135 / 0.4070 | 0.5314 / 0.0163 / 0.5022↑ | 0.6085 / 0.0186 / 0.5618↑ |
| Wine | 0.3779 / 0.1732 / 0.4782 | 0.4589 / 0.2199 / 0.5773↑ | 0.5330 / 0.2458 / 0.6541↑ |
| WordsSynonyms | 0.5406 / 0.3271 / 0.0977 | 0.6582 / 0.3953 / 0.1167↑ | 0.7572 / 0.4593 / 0.1338↑ |

# E    Robustness Analysis

In section $4.3.3$ (Robustness Analysis) of the main text, we explore the robustness of our model on one specific dataset (*SonyAIBORobotSurface*). Here we report the experimental results on all datasets to verify the point that our model is capable of correcting mistakes with the help of the term of $\mathcal{L}_{reconstruction}$ ($\mathcal{L}_{res}$) by recording the changes of the metrics.

As shown in Table 4, the experimental results are consistent with the Robustness Analysis of the main text. Three metrics in Table 4 from left to right are RI, NMI and ACC, respectively. The results in Table 4 can be divided into Group $A$ and $B$, consisting of three columns $a$, $b$, $c$ and $a$, $e$, $f$, respectively. **More intuitively, Groups $A$ and $B$ correspond to the results of the first and second rows in Figure 4 of the main text, respectively.** We also use arrows to indicate the changes in the performance. Within the Group $A$, from column $a$ to $b$, when we train DTCR with the loss term of shuffled K-means and the classification, the wrong clustering information does mislead the learning, decreasing the clustering performance. However, once putting the $\mathcal{L}_{res}$ term back (column $c$), the clustering performance is improved, indicating the learning of the model was corrected. Similarly, we do that again to check what happens without the $\mathcal{L}_{cls}$ loss term. Within the Group $B$, from column $a$ to $e$, even without the $\mathcal{L}_{cls}$ term, but with the help of the $\mathcal{L}_{res}$ term, the performance is still improved and less confused. Comparing column $b$ and $e$, it is clear that the $\mathcal{L}_{res}$ term enables our model to correct mistakes. Comparing column $c$ and $f$ shows that the earlier and longer the $\mathcal{L}_{res}$ term is used, the stronger the ability to prevent being misled by K-means and thus obtaining better performance.

Table 4: Robustness Analysis of DTCR

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

# F   Hyper-parameter Analysis

Here, we perform an empirical evaluation of the effect of the hyper-parameter $T$. We perform this evaluation with randomly chosen 10 datasets. We do the evaluation by varying one parameter at a time while maintaining the other parameters fixed (hidden size: $[100, 50, 50]$, $\lambda$: 1, dilation: $[1, 4, 16]$). As shown in Table 5, our method is robust to the hyper-parameter $T$.

Table 5: Hyper-parameter Analysis of $T$

| T<br>Dataset | 10 | | 20 | | 30 | | 40 | |
|---|---|---|---|---|---|---|---|---|
| | RI | NMI | RI | NMI | RI | NMI | RI | NMI |
| diatomsizeReduction | 0.9547 | 0.8801 | 0.9515 | 0.8874 | 0.9580 | 0.8910 | 0.9565 | 0.8901 |
| Mid.phal.outl.agegroup | 0.7966 | 0.4710 | 0.7967 | 0.4670 | 0.7969 | 0.4725 | 0.7981 | 0.4710 |
| Mid.phal.TW | 0.8635 | 0.4625 | 0.8632 | 0.4738 | 0.8630 | 0.4738 | 0.8636 | 0.4740 |
| OSULeaf | 0.7536 | 0.2859 | 0.7534 | 0.2532 | 0.7573 | 0.2398 | 0.7571 | 0.2640 |
| Plane | 0.9505 | 0.9296 | 0.9571 | 0.9295 | 0.9505 | 0.9296 | 0.9608 | 0.9296 |
| Prox.phal.outl.ageGroup | 0.7954 | 0.5377 | 0.7981 | 0.5596 | 0.7902 | 0.5491 | 0.8007 | 0.5628 |
| Prox.phal.TW | 0.8862 | 0.6299 | 0.8891 | 0.6342 | 0.8972 | 0.6517 | 0.8939 | 0.6571 |
| SonyAIBORobotSurfaceII | 0.8021 | 0.4813 | 0.8004 | 0.4767 | 0.8054 | 0.4864 | 0.8086 | 0.4932 |
| SwedishLeaf | 0.9088 | 0.5879 | 0.9074 | 0.5720 | 0.9111 | 0.5965 | 0.9094 | 0.5955 |
| WordsSynonyms | 0.8941 | 0.4287 | 0.8973 | 0.4402 | 0.8972 | 0.4300 | 0.8918 | 0.4180 |