[Reviews · NeurIPS 2019]

Reviewer 1



The paper is in general very well written, and it presents its contribution properly in relation to other results in the literature. While the proposed method is in the area of "deep learning", which means that there are no guarantees on the performance of the proposed method, an extensive simulation study shows that the contribution is promising.

Reviewer 2



The submission proposes a model for time-series clustering. The model is a novel combination of several existing components: a) a deep recurrent auto-encoder using dilated RNNs, b) a spectral relaxation of the K-means objective and c) a self-supervision loss to discriminate time-series corrupted by random shuffling from the original ones. The model is evaluated on a common benchmark for time-series clustering and achieves superior performance to existing methods. Overall I feel positive about the proposed method as the quantitative results look promising and using the spectral relaxation of K-means for deep clustering is novel and original. Nevertheless I do have some concerns about the submission in its current form: 1.) As far as I understand the time-series in the UCR benchmark are of fixed length for each category. In that sense there is no reason to explicitly model them as time-series but one could technically consider them as static vectors. Therefore one should compare the proposed method to an appropriate subset of the large body of recent work in deep learning clustering methods for static inputs (e.g. see [1] for recent overview). 2.) The submission has no discussion/analysis on how to treat the case if the number of clusters K is not known a priori, which is often the case for real world problems. 3.) The analysis and results in sections 4.3.2 and 4.3.3 are purely qualitative each evaluated on a single time-series clustering problem. It is not clear what general properties about the algorithm can be inferred. Minor comments - The paper mentions sequence-to-sequence modeling multiple times, it should reference the original paper using deep learning for seq2seq [2] - Typo in author name in line 119: Zhang et al. should be Zha et al. References [1] Aljalbout, E., Golkov, V., Siddiqui, Y., Strobel, M., & Cremers, D. (2018). Clustering with deep learning: Taxonomy and new methods. arXiv preprint arXiv:1801.07648. [2]Sutskever et al., Sequence to Sequence Learning with Neural Networks NIPS 2014 ——————————————————————————————— Given the author feedback, in particular the comparison with DEC and iDEC also on a separate, larger dataset, I am happy to increase my score to 7.

Reviewer 3



originality, quality, clarity, and significance. This paper proposes a method which integrates two new losses into an RNN autoencoder learning process for time series data. Specifically it adds a k-means loss to encourage k-means friendly clusters and a classification loss based on discriminating between shuffled input data and the true input data. It is evaluated on a number of standard benchmark datasets and shows an average improvement of 1% compared to the second best approach. There is a number of benefits to this work. a) I think the idea of using an RNN autoencoder, jointly trained with more cluster friendly losses, is good. Further, the 'fake data' discrimination process further strengthens this work. b) The evaluation process, while limited to benchmark datasets and (mostly) time-series algorithms, shows an improved performance, and the robustness of the approach. c) The ablation study helps clarify the individual contribution of each element of the proposed method. However, I think there are also a number of significant improvements that could be made to this work to bring it up to the level required for significance. a) The integration of k-means into an autoencoder loss function has been successfully used before in non time series specific methods. I would be interested in understanding the motivation behind the use of the k-means loss in this setting. How does this hold-up compared to dynamic time warping approaches? b) Some parameters are hardcoded, such as T and lambda at lines 137 and 154. How robust is the method to these? I think this is important as in the unsupervised setting typical supervised methods of choosing these parameters is unavailable. Can these parameters be set based on some other knowledge? c) On line 163 it is mentioned that datasets contain a train test split. What is the significance of saying this? In the unsupervised setting is the entire dataset not used for testing? d) I think each experiment should be run at least a few times (ideally 5 or 10) and the means and standard deviations reported. With approaches such as the proposed one it is difficult to understand how much the random initialization contributed to the final performance without this. e) The proposed method would also be much more convincing if larger more complex datasets were evaluated. f) I would find it useful to have a comparison with some other state of the art deep clustering methods (such as DEC[1], IDEC[2]). While not designed for time-series datasets, a comparison with them will improve both the clarity and potentially significance of the proposed method. g) The choice of metric is rather limiting. I would like to see accuracy and NMI also included as an evaluation metric as they are both commonly used in the literature and provide additional insights into the performance. Some minor suggestions to help improve the clarity of the paper Line 56 and 61 - 'a' should be 'an'. Line 192 - for / bracket placement. Line 185 - 'most best' is not grammatically correct, 'best' is fine. [1] Xie, Junyuan, Ross Girshick, and Ali Farhadi. "Unsupervised deep embedding for clustering analysis." International conference on machine learning. 2016. [2] Guo, Xifeng, et al. "Improved deep embedded clustering with local structure preservation." IJCAI. 2017.

[Author Response · NeurIPS 2019]

**Reply to Reviewer #1**

**Q1**: What other ways to generate fake sequences may be suitable for this problem?

A1: That is a good question. Although we here used a simple generation strategy (randomly shuffling some time steps), it has verified the improvement of the encoder's ability will boost the performance of clustering. In the future, we will consider to use GAN to generate some more difficult fake sequences to further improve the ability of the encoder.

**Reply to Reviewer #2**

**Q1**: Comparison with other state-of-the-art deep clustering methods which are not designed for time-series.

A1: Following your suggestion, we compare our method with two state-of-the-art deep clustering methods (i.e., DEC (Xie et al., 2016 ICML) and IDEC (Guo et al., 2017 IJCAI) ). The results of DEC and IDEC are obtained by running the original authors implementation code and we use the Rand Index (RI) to evaluate performance, consistent with our paper. As shown in Table 1, our method still achieves the best performance with the highest average RI of 0.7714. We will try to add these comparison results to the main text or the appendix, modulo the page limit.

Table 1: Comparisons on 36 time series datasets (The No. of datasets is consistent with the one in Table 2 in main text)

| Dataset | DEC(RI) | IDEC(RI) | DTCR(RI) | DTCR(NMI) | DTCR(ACC) | Dataset | DEC(RI) | IDEC(RI) | DTCR(RI) | DTCR(NMI) | DTCR(ACC) |
|---|---|---|---|---|---|---|---|---|---|---|---|
| 1 | 0.5817 | 0.6210 | **0.6868**(0.0026) | 0.5513(0.0022) | 0.6914(0.0028) | 19 | 0.5423 | 0.5423 | **0.5617**(0.0006) | 0.1150(0.0005) | 0.6733(0.0037) |
| 2 | 0.5954 | 0.6276 | **0.8046**(0.0018) | 0.5613(0.0013) | 0.5667(0.0013) | 20 | 0.8590 | 0.8626 | **0.8638**(0.0007) | 0.5503(0.0006) | 0.5940(0.0006) |
| 3 | 0.4947 | 0.6053 | **0.9000**(0.0001) | 0.7610(0.0001) | 0.8500(0.0001) | 21 | 0.7435 | 0.7324 | **0.7686**(0.0036) | 0.4094(0.0041) | 0.8395(0.0027) |
| 4 | 0.4737 | 0.4789 | **0.8105**(0.0033) | 0.5310(0.0035) | 0.8500(0.0031) | 22 | 0.7484 | 0.7607 | **0.7739**(0.0014) | 0.2599(0.0010) | 0.4430(0.0009) |
| 5 | 0.6859 | 0.6870 | **0.7501**(0.0022) | 0.5021(0.0020) | 0.6333(0.0017) | 23 | 0.9447 | 0.9447 | **0.9549**(0.0037) | 0.9296(0.0033) | 0.6286(0.0035) |
| 6 | 0.5348 | 0.5350 | **0.5357**(0.0011) | 0.0468(0.0010) | 0.8929(0.0015) | 24 | 0.4263 | **0.8091** | **0.8091**(0.0038) | 0.5734(0.0029) | 0.7805(0.0027) |
| 7 | 0.4921 | 0.5767 | **0.9286**(0.0016) | 0.8122(0.0015) | 0.9643(0.0016) | 25 | 0.8189 | **0.9030** | 0.9023(0.0023) | 0.6948(0.0017) | 0.6300(0.0017) |
| 8 | 0.9294 | 0.7347 | **0.9682**(0.0032) | 0.9418(0.0027) | 0.9118(0.0028) | 26 | 0.5732 | 0.6900 | **0.8769**(0.0033) | 0.7605(0.0027) | 0.9285(0.0021) |
| 9 | 0.7785 | 0.7786 | **0.7825**(0.0008) | 0.4553(0.0007) | 0.8050(0.0008) | 27 | 0.6514 | 0.6572 | **0.8354**(0.0016) | 0.6121(0.0017) | 0.9056(0.0016) |
| 10 | 0.5029 | 0.5330 | **0.6075**(0.0024) | 0.1180(0.0028) | 0.7083(0.0025) | 28 | 0.8837 | 0.8893 | **0.9223**(0.0021) | 0.6663(0.0019) | 0.3816(0.0018) |
| 11 | 0.6422 | 0.6233 | **0.6648**(0.0034) | 0.3691(0.0028) | 0.8000(0.0026) | 29 | 0.8841 | 0.8857 | **0.9168**(0.0022) | 0.8989(0.0018) | 0.6492(0.0018) |
| 12 | 0.5103 | 0.5114 | **0.9638**(0.0032) | 0.8056(0.0034) | 0.8525(0.0039) | 30 | 0.4984 | 0.5017 | **0.5659**(0.0006) | 0.3115(0.0008) | 0.5658(0.0013) |
| 13 | 0.4981 | 0.4974 | **0.6398**(0.0011) | 0.4200(0.0013) | 0.7867(0.0011) | 31 | 0.4991 | 0.4991 | **0.8286**(0.0028) | 0.3895(0.0033) | 0.9000(0.0023) |
| 14 | **0.5963** | 0.4956 | 0.5362(0.0035) | 0.0989(0.0036) | 0.6381(0.0031) | 32 | 0.6293 | 0.6338 | **0.6984**(0.0025) | 0.4713(0.0019) | 0.4253(0.0020) |
| 15 | 0.5099 | 0.5099 | **0.5759**(0.0017) | 0.2248(0.0016) | 0.7031(0.0015) | 33 | 0.5007 | 0.5016 | **0.7114**(0.0014) | 0.4614(0.0009) | 0.8622(0.0011) |
| 16 | 0.5311 | 0.5519 | **0.5913**(0.0016) | 0.2289(0.0014) | 0.7213(0.0017) | 34 | 0.5679 | 0.5597 | **0.7338**(0.0006) | 0.0228(0.0001) | 0.6775(0.0013) |
| 17 | 0.6475 | 0.6220 | **0.9763**(0.0016) | 0.9653(0.0009) | 0.9770(0.0017) | 35 | 0.4913 | 0.5157 | **0.6271**(0.0039) | 0.2887(0.0038) | 0.7963(0.0047) |
| 18 | 0.7059 | 0.6800 | **0.7982**(0.0028) | 0.4661(0.0017) | 0.7700(0.0023) | 36 | 0.8893 | 0.8947 | **0.8984**(0.0003) | 0.5448(0.0003) | 0.1630(0.0002) |

**Q2**: Discussion on how to use the presented method if the number of clusters K is not known a priori.

A2: For most of the clustering algorithms, the choice of hyper-parameter $K$ is indeed a tricky problem. We could use the popular "Elbow" method or Gap Statistic (Tibshirani et al., 2001 Royal Statistical Society) to choose $K$.

**Q3**: Whether the analysis and results in sections 4.3.2 and 4.3.3 are general properties of the algorithm?

A3: In fact, we conducted the experiments of sections 4.3.2 and 4.3.3 on all of the other datasets. The performance was consistent with the results in the article. We will add these results to the appendix.

**Reply to Reviewer #3**

**Q1**: The motivation behind the use of the k-means loss in this setting and how does this hold up compared to DTW?

A1: According to Zha et al. (2002 NIPS), K-means solves the minimum of a sum-of-squares cost function using coordinate descent, which is prone to local minima. Thus, they reformulate it as a trace maximization problem, which obtains optimal global solutions. Inspired by this, we reformulated K-means loss and integrated it into the autoencoder to guide representation learning. DTW may not be compatible with the trace maximization problem. This requires further investigation.

**Q2**: How robust is the method to hyper-parameters such as $T$ and $\lambda$?

A2: We performed experiments with $T \in [10, 20, 30, 40]$ to explore the impact of $T$ and verified our method is robust to $T$. We will add these results to the appendix. $\lambda$ determines the weight of K-means loss, which affects the performance to some extent. As mentioned in the main text, we used a grid search approach with $\lambda$ from [1, 1e-1, 1e-2, 1e-3].

**Q3**: Do datasets contain a train test split? Is the entire dataset used for testing?

A3: In the UCR time series benchmark, each data set has a default training and test set. To make a fair comparison, we adopted the same protocol used in USSL (Zhang et al., 2018 TPAMI), which is trained on the training set and evaluated on the test set.

**Q4**: The experiment should be run a few times to show the impact of the random initialization.

A4: Following your suggestion, we run each experiment 5 times and record means and standard deviations in Table 1. The values in parentheses present standard deviations, which are all less than 0.01. DTCR still achieves the best performance with the lowest average rank of 2.9583 and the highest average RI of 0.7714. We will update these results to the main text.

**Q5**: The proposed method would also be more convincing if larger more complex datasets were evaluated.

A5: Thank you for your suggestion. Following the YADING paper (Ding et al., 2015 VLDB), which was designed for large-scale time series clustering, a larger and more complex dataset (StarLightCurves: 9, 236 samples, each sample's length is 1, 024) is used to evaluate our method. We achieve NMI (normalized mutual information) metric of 0.6731, while NMI of YADING is 0.6000, NMI of DEC is 0.6058, and NMI of IDEC is 0.6056. Since USSL did not provide source code, we can not obtain the result of it for comparison. We will try to add the comparison to the main text or the appendix, modulo the page limit.

**Q6**: Comparison with other state-of-the-art deep clustering methods which are not designed for time-series.

A6: Please see Table 1. As the results shown, our method outperforms DEC as well as IDEC significantly.

**Q7**: Accuracy and NMI also should be included to evaluate the method performance.

A7: Due to the limitations of the layout and comparison methods results, we only show the results of RI. In fact, we also recorded NMI and accuracy. The results are shown in Table 1. And our method also achieve the lowest average rank of 2.1944 while USSL achieve 2.2361 on the NMI metric. We will add them to the text, modulo the page limit.

[Meta-Review · NeurIPS 2019]

This paper proposes a deep learning approach to clustering time series by combining a deep auto encoder and the spectral relaxation of K-means. The reviewers found the approach novel and the experimental evaluation of the approach reasonable. The concerns that the reviewers raised were addressed by the authors in their response. The authors should incorporate the suggestions that the reviewers provided to improve their paper for the camera-ready version.